# Functionalization of ZnAl-Layered Double Hydroxide with Ensulizole and Its Application as a UV-Protective Agent in a Transparent Polymer Coating

**DOI:** 10.3390/molecules28176262

**Published:** 2023-08-26

**Authors:** Klára Melánová, Kateřina Kopecká, Ludvík Beneš, Petr Kutálek, Petr Knotek, Zuzana Zmrhalová, Jan Svoboda

**Affiliations:** 1Joint Laboratory of Solid State Chemistry, Faculty of Chemical Technology, University of Pardubice, Studentská 84, 53210 Pardubice, Czech Republic; klara.melanova@upce.cz (K.M.); ludvik.benes@upce.cz (L.B.); petr.kutalek@upce.cz (P.K.); zuzana.zmrhalova@upce.cz (Z.Z.); 2SYNPO, akciová společnost, S. K. Neumanna 1316, 53207 Pardubice, Czech Republic; katerina.kopecka@synpo.cz; 3Department of General and Inorganic Chemistry, Faculty of Chemical Technology, University of Pardubice, Studentská 573, 53210 Pardubice, Czech Republic; petr.knotek@upce.cz; 4Institute of Organic Chemistry and Technology, Faculty of Chemical Technology, University of Pardubice, Studentská 573, 53210 Pardubice, Czech Republic

**Keywords:** layered double hydroxide, UV-protection, intercalation, ensulizole

## Abstract

In this study, we propose a promising photoprotective additive that combines the advantages of both organic UV absorbers and inorganic particles without compromising the properties of the paint material. This additive involves the intercalation of a well-known organic UV absorber, 2-phenylbenzimidazole-5-sulfonic acid (PBISA), into zinc-aluminum layered double hydroxide (ZnAl-LDH). Three ZnAl-LDH intercalates with PBISA were prepared using various methods based on either anion exchange or direct synthesis. The intercalates were characterized using powder X-ray diffraction, thermogravimetry, elemental analysis, and IR and UV-Vis spectroscopies. The composition and basal spacings of all three intercalates are very similar. An effective UV protection film was prepared when the ZnAl–PBISA–1 intercalate was incorporated into polyurethane-acrylate lacquer. The resultant UV protective film exhibited stability and uniform distribution of the intercalated fillers. Some minimal particle sedimentation and aggregation were observed on the cured film’s underside, but did not compromise the films’ UV protective properties. The prepared lacquers with intercalated fillers offer a viable solution for the surface modification of plastic products.

## 1. Introduction

Ultraviolet radiation can negatively affect the basic chemical bonds present in the structure of artificial materials, leading to the loss of primary properties from a change in color and structure, through loss of elasticity, to complete degradation and disintegration of the material. In order to preserve their desired properties, it is therefore necessary protect materials against the effects of UV radiation [1]. Currently, two basic methods of protection are used: (i) incorporation of UV-protecting molecules in the volume of the product, when stabilizers are mixed into the material directly during production, and (ii) surface protection, when the object is additionally surface-treated with a photoprotective coating system containing functional additives.

These UV-protective additives can be either organic compounds containing a chromophoric part capable of absorbing UV radiation or inorganic particles capable of absorbing the relevant wavelengths [2]. A disadvantage of organic molecules is their poor solubility in paint and coating systems and the related inhomogeneity of the distribution of the photoprotective effect. In addition, these molecules, due to their relatively low molecular weight compared to the molecular weights of the polymer chains forming the polymer matrix, can migrate to the surface of the photoprotective coating or potting resin and be washed away, leading to a decrease in efficiency accompanied by unwanted contamination of the environment.

The advantage of using inorganic particles, such include various forms of titanium dioxide or zinc oxide particles, is their ability to absorb a wider spectrum of wavelengths and their firm anchoring in the polymer matrix; on the other hand, the disadvantage is the relatively high concentration required to achieve comparable effectiveness with organic chromophoric molecules, which can have a negative effect on both the visual properties (color, gloss or transparency) and functional properties (flexibility, adhesion or strength) of the protective coating system or casting resin.

A way to eliminate the above-mentioned shortcomings can be a photoprotective additive combining the advantages related to the high efficiency of UV-absorbing molecules with the advantages of inorganic particles without negatively affecting the properties of the paint material. Such a photoprotective additive can be, for example, a layered double hydroxide (LDH) intercalated with a suitable organic UV absorber [3,4,5]. LDH may be represented by a general formula [M^II^_(1−*x*)_M^III^*_x_*(OH)_2_][*x*/nA^n−^]·mH_2_O, where M^II^ and M^III^ are divalent and trivalent cations, respectively, and A^n−^ represents exchangeable anions of charge n which compensates the positive charge induced by the presence of M^III^ in the layers. The anions are accommodated in the interlayer region, most often together with water molecules, and can be exchanged for other negatively charged species [6]. LDH intercalates with UV-adsorbing molecules have been studied for their application in sunscreen formulations [7]. 2-Phenylbenzimidazole-5-sulfonic acid (PBISA, Figure 1), commonly known as ensulizole or Eusolex 232, is an effective and recommended sunscreen agent and it is a part of many commercially available sunscreen products [8,9]. ZnAl-LDH intercalated with PBISA was prepared either via anionic exchange from the chloride form of ZnAl-LDH [10] or via direct coprecipitation [11] and tested for use in sunscreen formulations. These experiments showed that both intercalation products maintained the sunscreen’s properties, indicating that LDHs are suitable matrices for sunscreen formulations. However, to the best of our knowledge, such an intercalate has not yet been used to protect polymers against UV radiation.

In this paper, we describe other ways of preparing PBISA intercalates and their use as a filler for polyurethane-acrylate lacquer.

## 2. Results

Three different methods were used for the intercalate preparation. Two of them are based on anionic exchange. In the first case, PBISA was intercalated into the pre-prepared ZnAl–CO_3_; in the second case, ZnAl–CO_3_ was first delaminated using a high-shear homogenizer to produce particles with a thickness of several layers. Although the carbonate ions are strongly held in the interlayer region of starting ZnAl–CO_3_, they can be replaced using 2-phenylbenzimidazole-5-sulfonic acid, where the reaction is favored by removing the carbonate ions from the solution in the form of CO_2_. Another advantage of this procedure is that it is not necessary to work in an inert atmosphere and to use CO_2_-free water. The last method was the direct reaction of zinc oxide with aluminum nitrate, PBISA, and urea solution.

All three intercalates are white crystalline solids, and their powder diffraction patterns contain several sharp basal reflections (see Figure 2). Their basal spacings are very close to each other, slightly higher than the value 21.0 Å given by Mohsin [11] and similar to the value 21.9 Å given in the work of Perioli [10]. The difference in the basal distance between our samples and those from the literature may be due to differences in the Zn and Al contents (Zn_0.66_Al_0.34_ in [10] and Zn_0.69_Al_0.31_ in [11]), which may lead to slightly different amounts of intercalated PBISA anions (0.285 PBISA together with 0.055 Cl per formula unit in [10] and 0.31 PBISA per formula unit in [11]) and different amounts of co-intercalated water (0.9 water per formula unit in [10] and 1.06 of PBISA water per formula unit in [11]).

The composition of the intercalates prepared was determined using EDX, elemental and thermogravimetric analyses. The formulas of the compounds prepared are given in Table 1, and the results of the analyses are summarized in Appendix A. As can be seen from the table, the composition of all three intercalates is very similar and is comparable to the intercalate obtained in Perioli’s work [10]. For both samples prepared via intercalation into ZnAl–CO_3_, the Zn/(Zn + Al) ratio remained the same as in the parent ZnAl–CO_3_. The anion content is slightly higher in the ZnAl–PBISA–3 sample, which was prepared from a delaminated host. In addition, the reaction time was slightly shorter with this procedure.

SEM images of the intercalates prepared are shown in Figure 3. The parent ZnAl–CO_3_ has a morphology typical of the LDH, with nanometer-thin plate-like particles, whose diameters range from 5 μm to 10 μm. The intercalates exhibit agglomerates of thin plate-like particles. The size of the agglomerates in ZnAl–PBISA–1 and ZnAl–PBISA–3 is comparable, although ZnAl–PBISA–3 was prepared from delaminated ZnAl–CO_3_. Thus, agglomeration of the particles probably occurred during functionalization. The largest agglomerates were observed in the sample ZnAl–PBISA–2.

All of the intercalates decompose in three steps (see Figure 4): the first one corresponds to the interlayer water release, the second one to the release of water formed by condensation of hydroxyl groups and CO_2_ from the carbonate decomposition, and the third one to the decomposition of intercalated PBISA anions. The found and calculated weight loss values are given in Appendix A. The end product of the thermal decomposition is a mixture of ZnO (JCPDS No. 04-013-7122) and ZnAl_2_O_4_ (JCPDS No. 01-074-1136) [12] in the ratio corresponding to the Zn/Al ratio in the intercalate. As an example, the powder X-ray pattern of the ZnAl–PBISA–3 decomposition product is shown in Appendix A.

Infrared spectra of all the intercalates, together with the spectra of pure PBISA and its sodium salt, are shown in Figure 5. It is very difficult to assign individual bands to the corresponding vibrations because the regions of occurrence of the bands corresponding to the vibrations of the aromatic part of the molecule overlap with the region of occurrence of the bands corresponding to the vibrations of the sulfonic group [13]. Looking at the intercalate spectrum as a whole, we can see that it is more similar to the salt spectrum, which is consistent with the idea of PBISA anions intercalated in the LDH interlayer space. The bands at about 1086 and 1029 cm^−1^ are probably caused by C=N or C=C vibration because their positions and intensity ratio are nearly the same in pure acid and in its salt. The positions of the bands at 631, 781, 950, 1319, and 1458 cm^−1^ do not differ much. On the other hand, the two bands in the acid spectrum at 1176 cm^−1^ and 1229 cm^−1^ change to a broad band with a maximum at about 1170 cm^−1^ with a more or less distinct shoulder at about 1220 cm^−1^ in both the salt and intercalate spectra. The bands at 1498, 1514, and 1568 cm^−1^ observable in the PBISA spectra are not present in the spectra of the salt and the intercalates, but a band at 1540 cm^−1^ appears there. A sharp band at 1634 cm^−1^ present in the PBISA spectrum is significantly broadened in the intercalates’ spectra, which may be caused by the deformation vibration of co-intercalated water molecules. The band at about 1360 cm^−1^ corresponding to the vibrations of the intercalated carbonate anions is not observed in the spectra of the intercalates, which confirms the almost complete exchange of carbonate anions for PBISA anions. At higher wavenumbers, only a very broad band with a maximum of about 3500 cm^−1^ is present in the spectra of all intercalates, which is probably caused by hydroxy groups of the LDH (see the whole spectrum in Appendix A).

As was shown in [14], the UV-absorption spectra of PBISA are strongly influenced by the solvent used and the pH of the solution. UV-Vis spectra of the intercalates and pure PBISA measured in a mixture with Al_2_O_3_ are shown in Figure 6. Pure PBISA has a band with the highest intensity at 261 nm and three other intensive bands at 290, 316, and 330 nm. In the spectra of the intercalates, a very broad band with a maximum at 313–318 nm with a more or less distinct shoulder at about 329 nm can be observed.

For testing the compatibility of the intercalate with a polyurethane-acrylate lacquer, a series of samples containing ZnAl–PBISA–1 and, for comparison, ZnAl–CO_3_ and PBISA alone were prepared. Polyurethane-acrylate lacquer LV CC 220 retained good wetting and leveling properties even when fillers were used. The films prepared were homogeneous in thickness and flexible. The distribution of the fillers in the system was evaluated using electron microscopy (see Figure 7). The ZnAl–CO_3_ and ZnAl–PBISA–1 filler particles were distributed homogeneously in the area; however, when the film dried, they sedimented and the lower side of the film contained more particles. This is not a problem if the system is used as a UV protective coating. In the case of incorporating PBISA itself, the distribution of particles on the surface is inhomogeneous. Clusters of particles are formed in some places, while in other places, the particles are not detectable at all.

To measure the optical transmittance, foil samples were placed on the detector to minimize the effect of scattering on solid particles of the sample. Measurements were performed for five different locations of the film. The optical transmittance of the samples varied significantly (see Figure 8). All spectra are affected by the optical inhomogeneities typical for modified polyurethane films. The modification led to an increase in the photon multiple scattering and consequently to a decrease in the optical transmission in the spectral region with a low absorption coefficient, which could be caused by the polymeric [15], organic [16] or inorganic [17] fillers. The highest optical transparency was shown by the standard foil without a filler (polyurethane-acrylate co-polymer) and the film filled with ZnAl–CO_3_ (low dimensional layered materials with limited photon scattering). The samples with ZnAl–PBISA–1 and pure PBISA had a more significant scattering due to the increase in the size of the optically inhomogeneous parts—in our case, the filler (see SEM images in Figure 7). A hint of an absorption band around 320 nm (see the spectra of the fillers at Figure 6) was detectable on the sample with ZnAl–PBISA–1.

To compare the transmittance at individual wavelengths, transmittance averages between individual samples were compared. A statistical box plot is shown for 320 nm, corresponding to the border of UV-A and UV-B regions, where there is an absorption band of the fillers (Figure 9). It follows from the measured data that the sample with ZnAl–PBISA–1 has the highest absorption of radiation, i.e.., lowest transmittance, at the wavelength of 320 nm. The transmittance is at the level of one-tenth that of the standard foil without the filler. It is also worth noting the small deviations from the mean value, which indicates that the film properties are practically the same at all measured locations, which is consistent with the observed homogeneous distribution of the filler in the system. The developed UV absorber therefore exhibits higher radiation absorption/shielding compared to the organic molecule PBISA itself and, at the same time, significantly improves its compatibility with polyurethane-acrylate lacquer LV CC 220.

## 3. Materials and Methods

Ensulizole (2-phenylbenzimidazole-5-sulfonic acid, PBISA or Eusolex 232) was obtained from the commercial provider Merck s.r.o., Prague, Czech Republic, as well as other commercially available chemicals not further specified. Polyurethane-acrylate lacquer LV CC 220 and hardener LV BU 45 were provided by SYNPO, akciová společnost, Pardubice, Czech Republic.

### 3.1. Syntheses

Layered double hydroxide with the formula Zn_0.61_Al_0.39_(OH)_2_](CO_3_)_0.165_·0.5H_2_O (denoted further as ZnAl–CO_3_) was prepared via a urea method [18,19]. Intercalates with PBISA were prepared in three different ways:

ZnAl–PBISA–1: ZnAl–CO_3_ (2.5 g) was refluxed with a solution of PBISA (3.2 g) in a mixture of ethanol (100 mL) and water (300 mL) overnight.

ZnAl–PBISA–2: ZnO (1.07 g) was mixed with a solution of PBISA (1.81 g) in ethanol (100 mL) and refluxed for 4 h. A solution of Al(NO_3_)_3_·9H_2_O (2.54 g) and urea (3.92 g) in water (300 mL) was then added and the reaction mixture was refluxed overnight.

ZnAl–PBISA–3: ZnAl–CO_3_ (2.0 g) was suspended in isopropyl alcohol (400 mL) and delaminated using a high-shear homogenizer IKA T10 Standard Ultraturax^®^ (IKA^®^-Werke GmbH & Co KG, Staufen, Germany) for 5 min. An aqueous solution of PBISA (2.6 g, 200 mL) was added and the reaction mixture was refluxed for 8 h.

Solid phases were separated by means of centrifugation, washed three times with ethanol, and dried in air.

### 3.2. Preparation of Polymer Films

The appropriate amount of microparticle filler was mixed into 30 g of polyurethane-acrylate varnish LV CC 220 using a dispersing disc (see Table 2). The mixture was stirred at 600 rpm for 1 h. Then, the mixture was left to stand until the next day when 15 g of hardener LV BU 45 N was manually mixed with a stick, and samples of free films were prepared using a gap-applicator with a 400 µm slot on a polypropylene substrate. The samples were allowed to dry at room temperature for seven days.

### 3.3. Instrumentation

X-ray diffraction analysis. Powder X-ray diffraction data were obtained with a D8-Advance diffractometer (Bruker AXS, Karlsruhe, Germany) with Bragg–Brentano *θ*-*θ* geometry (40 kV, 30 mA) using CuKα radiation with a secondary graphite monochromator. The diffraction angles were measured at room temperature from 2 to 65° (2*θ*) in 0.02° steps with a counting time of 10 s per step.

Thermogravimetric analysis. The thermogravimetric analysis was performed using a homemade apparatus constructed of a computer-controlled oven and a Sartorius BP210 S balance (Göttingen, Germany). The measurements were carried out in air between 30 and 960 °C at a heating rate of 5 °C min^−1^.

Scanning electron microscopy and energy-dispersive X-ray analysis (EDX) were performed using an electron scanning microscope JEOL JSM-5500LV (Jamagata, Yamaguchi, Japan) equipped with an energy-dispersive X-ray microanalyzer from IXRF Systems (detector GRESHAM Sirius 10). The accelerating voltage of the primary electron beam was 20 kV.

Organic elemental analysis (C, H, N) was performed on a Flash 2000 CHNS Elemental Analyzer (Thermo Fisher Scientific, Milan, Italy).

Infrared spectra. Infrared spectra in the range of 400–4000 cm^−1^ were recorded at 64 scans per spectrum at 2 cm^−1^ resolution using a fully computerized Thermo Nicolet NEXUS 870 FTIR (Madison, WI, USA) Spectrometer equipped with a DTGS TEC detector. Measurements of the powdered samples were performed ex situ in the transmission mode in KBr pellets. All spectra were corrected for the presence of moisture and carbon dioxide in the optical path.

UV-Vis spectra. Diffuse reflectance was measured using a CINTRA2020 UV-Vis spectrophotometer (GBC Scientific Equipment, Keysborough, Australia). SiO_2_ cuvettes with an optical path length of 2 mm and pure Al_2_O_3_ as the background were used. The samples were diluted with Al_2_O_3_ in a ratio of 1:50. Spectra were measured in the wavelength range from 800 to 200 nm. The reflectance was converted into the dependence of the Kubelka–Munk function on energy according to the equation F(R_∞_) = (1 − R_∞_)^2^/2R_∞_, where R_∞_ corresponds to the diffuse reflectance on a semi-infinite layer. In addition, the spectra were normalized (range 0, 1). 

## 4. Conclusions

Three ZnAl-layered double hydroxide intercalates with ensulizole were prepared via different methods. The intercalates were characterized using powder X-ray diffraction, thermogravimetric analysis, scanning electron microscopy, energy-dispersive X-ray microanalysis, organic elemental microanalysis, infrared spectroscopy, and UV-Vis absorption in the solid state. The composition of the intercalates was suggested based on these methods.

The ZnAl–PBISA–1 intercalate showed significant UV protection when applied in polymeric lacquer. The UV protective film prepared was stable and had a homogeneous distribution of the intercalate filler. A slight tendency for particle sedimentation and aggregation was evident on the underside of the cured film. The distribution of these particles was homogeneous in all cases and did not affect the UV-protective properties of the films.

It can be assumed that the prepared varnishes with intercalates as fillers can be advantageously used for the surface treatment of plastic products. The newly designed varnishes show similar properties to the commercial product LV CC 220. Still, the proven reduced UV transmittance can significantly help to protect the materials coated with them against photochemical corrosion. This can subsequently lead to an extension of the service life of the coated materials.

## Figures and Tables

**Figure 1 molecules-28-06262-f001:**
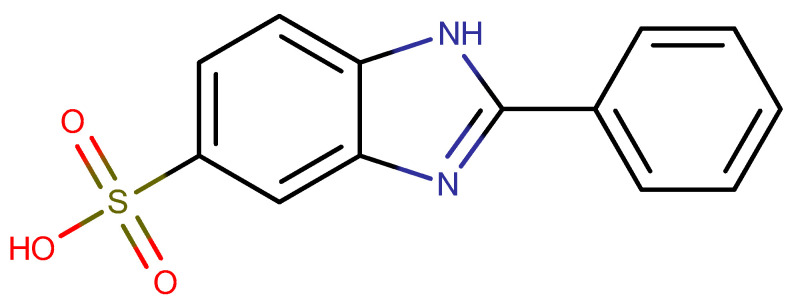
Structural formula of 2-phenyl-1*H*-benzo[*d*]imidazole-5-sulfonic acid (PBISA).

**Figure 2 molecules-28-06262-f002:**
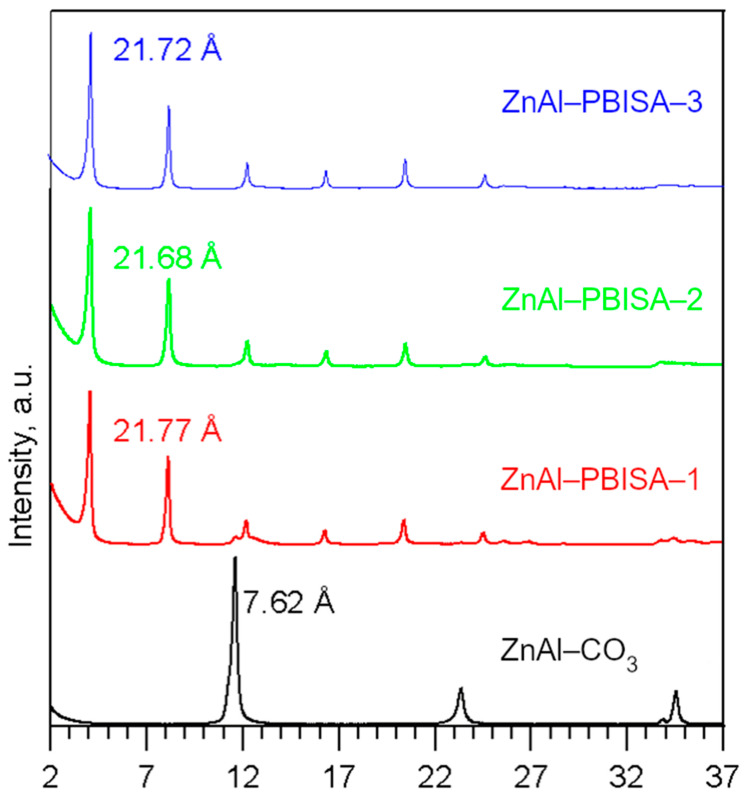
Powder X-ray patterns of the intercalates prepared.

**Figure 3 molecules-28-06262-f003:**
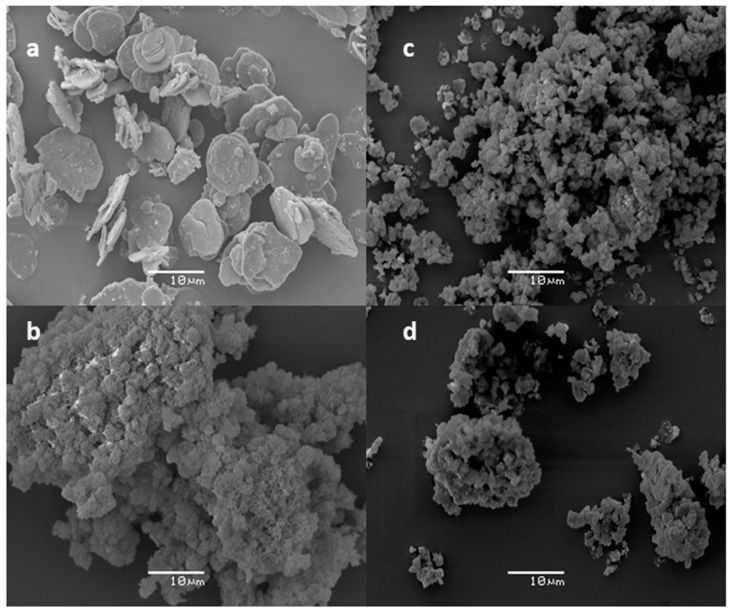
SEM images of parent ZnAl–CO_3_ (**a**), ZnAl–PBISA–2 (**b**), ZnAl–PBISA–1 (**c**), and ZnAl–PBISA–3 (**d**).

**Figure 4 molecules-28-06262-f004:**
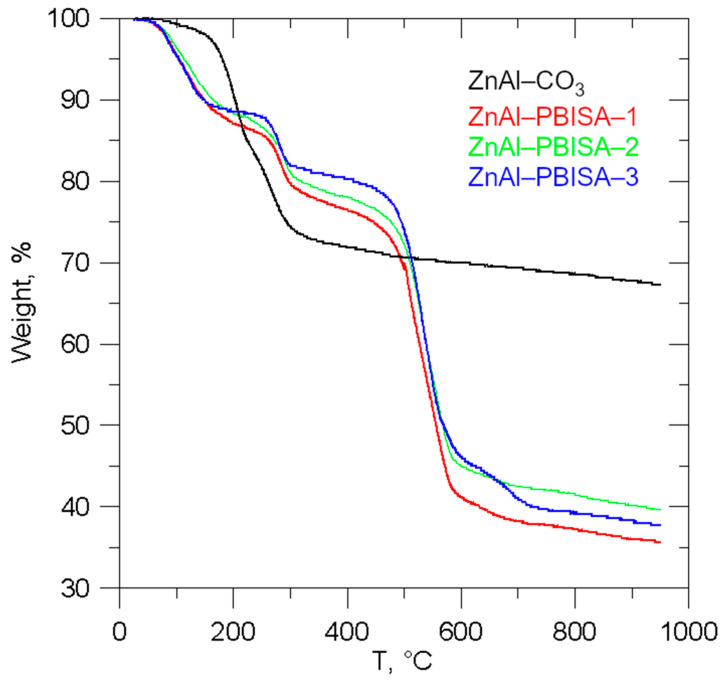
Thermogravimetric curves of the intercalates prepared.

**Figure 5 molecules-28-06262-f005:**
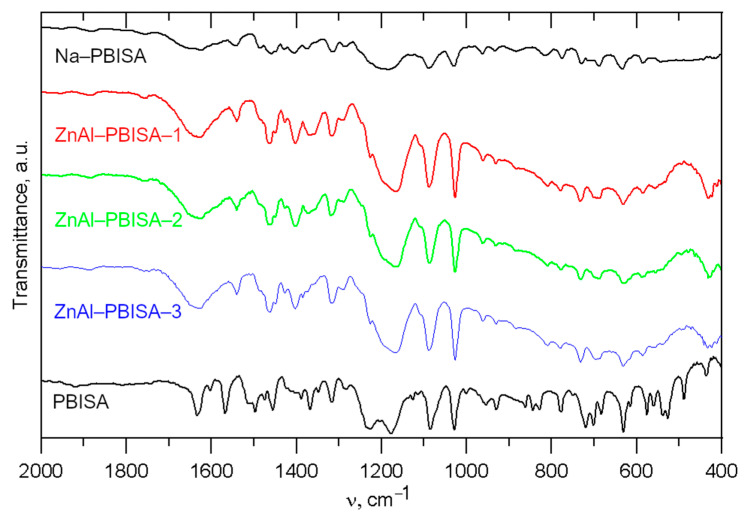
Infrared spectra of the intercalates prepared together with IR spectra of pure PBISA and its sodium salt.

**Figure 6 molecules-28-06262-f006:**
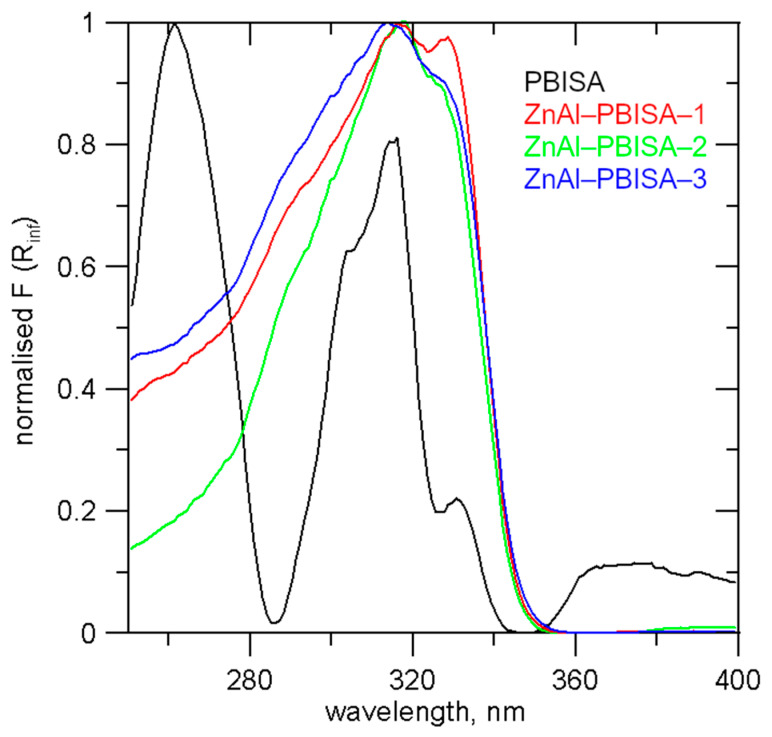
UV-Vis spectra of the intercalates and PBISA.

**Figure 7 molecules-28-06262-f007:**
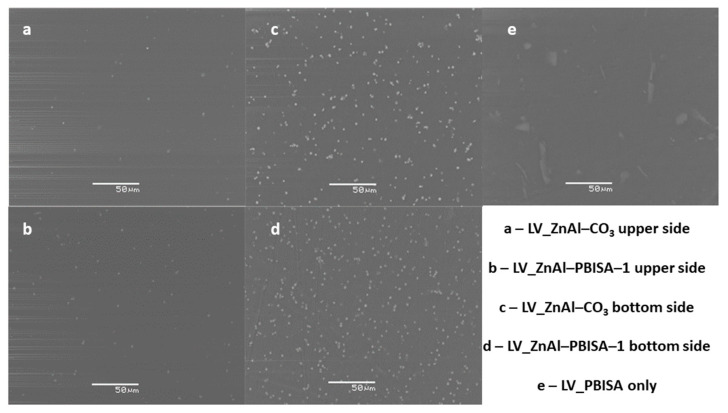
SEM images of polyurethane-acrylate lacquer film LVCC 220 with the appropriate fillers.

**Figure 8 molecules-28-06262-f008:**
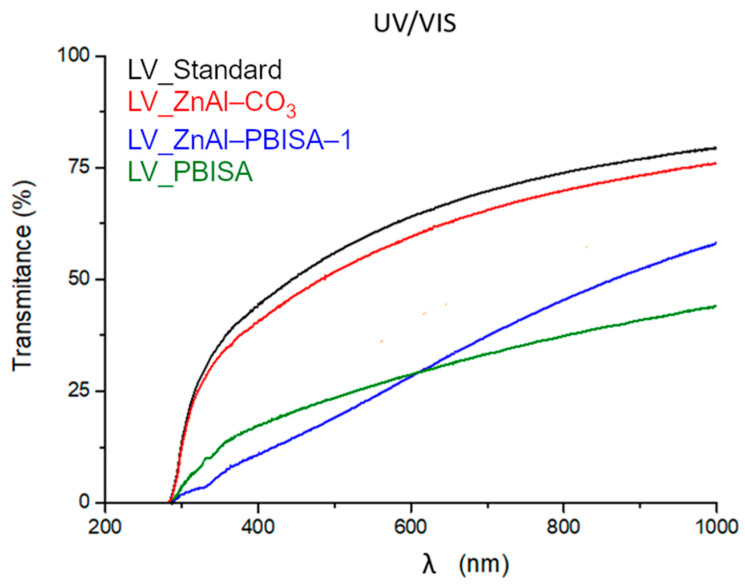
Spectral dependence of the optical transmittance of filled films of polyurethane-acrylate lacquer.

**Figure 9 molecules-28-06262-f009:**
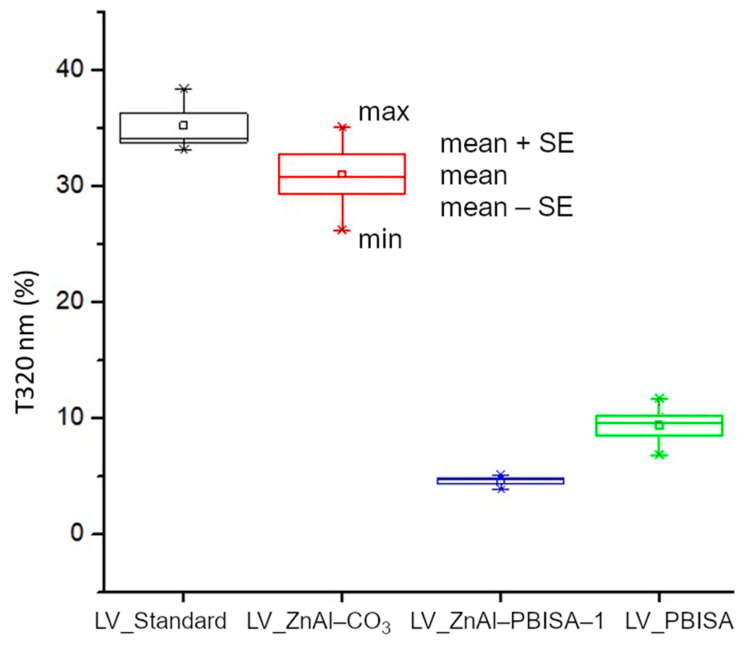
Comparison of optical transmittance of foils at 320 nm.

**Table 1 molecules-28-06262-t001:** Composition of the intercalates prepared.

Sample	Zn/(Zn + Al)	S/Al	Formula
ZnAl–PBISA–1	0.61	0.68	Zn_0.61_Al_0.39_(OH)_2_(C_13_H_9_N_2_O_3_S)_0.27_(CO_3_)_0.06_·1.2H_2_O
ZnAl–PBISA–2	0.60	0.56	Zn_0.60_Al_0.40_(OH)_2_(C_13_H_9_N_2_O_3_S)_0.22_(CO_3_)_0.09_·1.2H_2_O
ZnAl–PBISA–3	0.61	0.74	Zn_0.61_Al_0.39_(OH)_2_(C_13_H_9_N_2_O_3_S)_0.29_(CO_3_)_0.05_·1.2H_2_O

**Table 2 molecules-28-06262-t002:** Preparation of the films.

Designation	LV_Standard	LV_ZnAl–CO_3_	LV_ZnAl–PBISA–1	LV_PBISA
LV CC 220	30 g	30 g	30 g	30 g
ZnAl–CO_3_	—	0.23 g	—	—
ZnAl–PBISA–1	—	—	0.41 g	—
PBISA	—	—	—	0.18 g
LV BU 45 N	15 g	15 g	15 g	15 g

## Data Availability

Not applicable.

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
