# Peer review of "Functionalization of ZnAl-Layered Double Hydroxide with Ensulizole and Its Application as a UV-Protective Agent in a Transparent Polymer Coating"

_molecules, 2023, doi:10.3390/molecules28176262_

Round 1
Reviewer 1 Report
In this manuscript, three different membranes were synthesized by three different methods. The combination between ZnAl-LDH and PBISA forms a coating compound that shows excellent UV protection function in transparent polymer coatings. ZnAl-PBISA1 was mixed with polyurethane acrylic paint to form a UV protective film. The synthesized UV-protective film has excellent stability and uniform distribution of interlayer fillers. By testing the film, it was found that the film has higher radiation absorption and lower light transmittance compared to films made with pure PBISA and ZnAlCO3. The full text has clear logic, fluent language, and detailed content, but there are still some issues that need improvement. The following are the suggestions for revision:
1. What is the main difference in these three samples obtained by different synthetic methods? The authors should clearly clarify this point.
2. ICP may be required to verify the Zn/Al ratio in these three samples.
3. It seems that the amount of PBISA has an influence on the UV protection function of the materials. Is it possible for the author to just adjust the proportion of PBISA? And, is it still works better by use using the physical mixture between ZnAl-LDH and PBISA.
4. This article lists three ZnAl-PBISA compounds and conducts basic characterization, but the latter half only tests PBISA1 and concludes that ZnAl-PBISA1 has better performance, without comparing the performance of ZnAl-PBISA2/3.
5. In lines 143 to 144, it is mentioned that there is a very wide band at 3500 cm-1 at higher wavenumbers, but there is no data to support this.
6. In lines 171-177, the text states that the measurements were taken at five different positions of the membrane, which are not shown in the figure.
7. The conclusion states that "paint prepared with embedded fillers can be effectively used for surface modification of plastic products, preventing photochemical corrosion and prolonging their service life", without experimental data support.
Author Response
Reviewer 1
In this manuscript, three different membranes were synthesized by three different methods. The combination between ZnAl-LDH and PBISA forms a coating compound that shows excellent UV protection function in transparent polymer coatings. ZnAl-PBISA1 was mixed with polyurethane acrylic paint to form a UV protective film. The synthesized UV-protective film has excellent stability and uniform distribution of interlayer fillers. By testing the film, it was found that the film has higher radiation absorption and lower light transmittance compared to films made with pure PBISA and ZnAlCO3. The full text has clear logic, fluent language, and detailed content, but there are still some issues that need improvement. The following are the suggestions for revision:
Thanks to the opponent for all suggestions mentioned.
- What is the main difference in these three samples obtained by different synthetic methods? The authors should clearly clarify this point.
Thanks for the question. The important thing is that with all the methods used, we prepare essentially the same material, differing only little in the tendency to aggregate; see Figure 2. For this reason, we chose the ZnAl-PBISA1 intercalate for compatibility testing with polyurethane varnish.
- ICP may be required to verify the Zn/Al ratio in these three samples.
Thank you for the recommendation. We determined the Zn/Al ratio using a less precise method of energy dispersive microanalysis. Because the ratio of these elements in the products (intercalates) corresponded to the ratio in the starting substances, this more accurate method of determination was not used. The crystalline fraction in the residue after thermogravimetric analysis was analyzed by X-ray powder diffraction. The Zn/Al molar ratio calculated based on corundum numbers also corresponded to the given composition; see the supplementary material. We have to order the ICP analysis as a commercial service, and therefore, mainly for reasons of time, we have decided not to list it here yet. If this analysis is required despite all the facts mentioned above, we will need more time to perform the relevant experiments.
- It seems that the amount of PBISA has an influence on the UV protection function of the materials. Is it possible for the author to just adjust the proportion of PBISA? And, is it still works better by use using the physical mixture between ZnAl-LDH and PBISA.
The amount of PBISA has an effect on UV protection. The amount of PBISA in the prepared intercalates is determined by their host-guest properties. The experiments show that PBISA alone does not have a good distribution in the film (see. Figures 6 and 8) and is not well-compatible with the polyurethane used. The host itself does not have such great UV protection capabilities. The use of prepared intercalates combines both advantages. Experiments with physical mixtures were not performed. We assume that the desired synergistic effect mentioned above would not occur.
- This article lists three ZnAl-PBISA compounds and conducts basic characterization, but the latter half only tests PBISA1 and concludes that ZnAl-PBISA1 has better performance, without comparing the performance of ZnAl-PBISA2/3.
The opponent is right that it would be good to test all the prepared intercalates in the resulting films and compare them with each other. The presented work is based on a technological assignment, where one part of the project examined the compatibility of polyurethane films with additives, and the other part dealt with the preparation of the additive. All three prepared intercalates are practically identical, and therefore only one was selected as a filler for testing. When we talk about the better performance of ZnAl-PBISA1 in the films, we mean the comparison with the host itself or PBISA itself, not the comparison of the intercalates with each other. Extending the work to include the preparation and testing of two more films is impossible due to the dependence on the commercial partner, who considers the project to be closed.
- In lines 143 to 144, it is mentioned that there is a very wide band at 3500 cm-1 at higher wavenumbers, but there is no data to support this.
Thanks for the advice. The relevant full-range spectrum was inserted into the supplementary, and the text was adequately modified.
- In lines 171-177, the text states that the measurements were taken at five different positions of the membrane, which are not shown in the figure.
Thanks for the improvement idea. SEM images (Figure 6) are representative images documenting the typical appearance of the top and bottom surfaces of the films and cannot be considered as the exact selection on which optical transmittance measurements were provided. The film was randomly applied to the detector five times, and the measured transmittance was recorded. These measurements were then statistically evaluated, see Figure 8.
- The conclusion states that "paint prepared with embedded fillers can be effectively used for surface modification of plastic products, preventing photochemical corrosion and prolonging their service life", without experimental data support.
Thank you for the correction. The text was modified respectively.
Reviewer 2 Report
In this manuscript, Svoboda and co-workers report on a new method for intercalating an organic UV-absorber (ensulizole) into ZnAl-layered double hydroxides (LDH) through anion exchange from carbonate to develop new hybrid UV-absorbers. Although almost same structures have already been reported (e.g. J. Phys. Chem. Solids, 2006, 67, 1079) as sited on ref. 10, there appear to be some novelties in this manuscript, including the methodology (exchange from carbonate) and its use as fillers. In addition, the UV transmittance of the hybrid materials was examined and found to perform better than the original UV-absorber (ensulizole). Therefore, I believe this manuscript is appropriate for publication in this journal with minor technical revision as follows.
1) It would be more helpful to reader if the structural formula of ensulizole was shown in this manuscript.
2) I believe that it is preferable to have some discussion of the changes in interlayer distances by intercalating the UV-absorber based on the PXRD results shown in Figure 1.
3) Does the third weight loss in TG (Figure 3) correspond to the amount of interlayer guest molecules estimated from other analyses?
4) In Figure 7, why does the UV transmittance of the materials decrease significantly even in the higher wavelength region (> 600 nm) where the materials have no absorbance? I believe it is desirable to discuss this issue somewhere in this manuscript.
5) In Figure 8, why is the transmittance of this material higher than that of the original compound? I believe it is desirable to discuss this issue in terms of the hybridized structure somewhere in this manuscript.
Author Response
Reviewer 2
In this manuscript, Svoboda and co-workers report on a new method for intercalating an organic UV-absorber (ensulizole) into ZnAl-layered double hydroxides (LDH) through anion exchange from carbonate to develop new hybrid UV-absorbers. Although almost same structures have already been reported (e.g. J. Phys. Chem. Solids, 2006, 67, 1079) as sited on ref. 10, there appear to be some novelties in this manuscript, including the methodology (exchange from carbonate) and its use as fillers. In addition, the UV transmittance of the hybrid materials was examined and found to perform better than the original UV-absorber (ensulizole). Therefore, I believe this manuscript is appropriate for publication in this journal with minor technical revision as follows.
Thanks to the opponent for all suggestions mentioned.
- It would be more helpful to reader if the structural formula of ensulizole was shown in this manuscript.
Thank you for the suggestion. The structural formula was inserted into the manuscript and the text was appropriately rewritten.
- I believe that it is preferable to have some discussion of the changes in interlayer distances by intercalating the UV-absorber based on the PXRD results shown in Figure 1.
We agree with the opponent. The discussion of changes in powder diffractograms of individual intercalates has been revised and enlarged.
- Does the third weight loss in TG (Figure 3) correspond to the amount of interlayer guest molecules estimated from other analyses?
Thanks for the question. In the supplementary, we added the calculation of theoretical mass losses for thermogravimetric analysis and their comparison with measured values. See Table S2 in the supplementary. The crystalline residue from the thermogravimetric analysis was also analyzed by powder X-ray diffraction. ZnO and ZnAl2O4 were detected. Furthermore, their proportional representation was determined. See the supplementary.
- In Figure 7, why does the UV transmittance of the materials decrease significantly even in the higher wavelength region (> 600 nm) where the materials have no absorbance? I believe it is desirable to discuss this issue somewhere in this manuscript.
Thank you for your comments, the manuscript was expended for the description of the spectroscopic behavior: The optical transmittance of the samples varied significantly (see Figure 8). All spectra are affected by the optical inhomogeneities typical for the modified polyurethane films. The modification led to the increase of the photon multiple scattering and consequently to the decrease of the optical transmission in the spectral region with low absorption coefficient and could be caused by the polymeric (DOI:10.1155/2014/487343), organic (DOI:10.1007/s11998-021-00492-y) or inorganic (DOI:10.1081/DIS-200025688) grafting/additives/fillers. The highest optical transparency was shown by the standard foil without a filler (polyurethane-acrylate co-polymer) and the film filled with ZnAl-CO3 (low dimensional layered materials with limited photon scattering). The samples with ZnAl-PBISA1 and pure PBISA had a more significant scattering due to the increase of the size of the optically inhomogeneous parts, in our case, the filler (see SEM images in Fig. 7). A hint of an absorption band around 320 nm (see the spectra of the fillers at Fig. 6) was detectable on the sample with ZnAl-PBISA1.
- In Figure 8, why is the transmittance of this material higher than that of the original compound? I believe it is desirable to discuss this issue in terms of the hybridized structure somewhere in this manuscript.
Sorry, but we're not sure what the reviewer means by this question. If it is i) a comparison of the optical transmittance values for the ZnAl-PBISA1 filler (now Figure 8 and 9), the authors are aware that the values of the transmittance at 320 nm could not be directly compared to the normalized values of DR-UV-Vis spectra (Fig. 6) due to the differences in the experimental set-up (diffuse reflectance of powder and direct transmission) as well as mathematical models (Kubelka-Munk eq. and direct experimental values for DR and transmission, respectively).
If it is ii) the comparison of statistical errors in Figure 9, we suppose that Figure 9 compares the average transmittance of sample films at a wavelength of 320 nm. A statistical box plot is shown taking into account the minimum (0th percentile), maximum (100th percentile), mean, and corresponding standard deviation. The original film has the highest transmittance which corresponds with the highest statistical box plot in the gaff.
Round 2
Reviewer 1 Report
The performance of the physical mixture between ZnAl-LDH and PBISA should be measured.
Author Response
Reviewer 1
The performance of the physical mixture between ZnAl-LDH and PBISA should be measured.
We agree with the opponent that preparing and measuring a film of a physical mixture of BPISA and ZnAl-CO3 would be interesting and expedient. During the preparation of the LV_PBISA film, an incompatibility was observed during processing (mentioned in the manuscript), which we also assumed during the preparation of the mixture under consideration. For this reason, the preparation of the physical mixture was not carried out. We will certainly try to avoid this mistake in future experiments. However, the possible preparation of this sample for measurement in the foreseeable future is unavailable to us, similar to the preparation of the other two films with intercalated BPISA2 and PBISA3. In the manuscript, the compatibility of the ZnAl-PBISA1 intercalate with the LV_Standard polymer in the film and the minimum level of optical transmittance at 320 nm was an interesting result for us. This finding was also interesting for an industrial partner in preparing a commercial product.